# Women, Entrepreneurship, and Sustainability: The Case of Saudi Arabia

Nadia A. Abdelmegeed Abdelwahed [1,*] , Bettina Lynda Bastian [2] and Bronwyn P. Wood [3,*]

1    Department of Business Management, College of Business Administration, King Faisal University, Al Hofuf 31982, Saudi Arabia
2    College of Business and Financial Services (CBFS), Royal University for Women, Riffa P.O. Box 37400, Bahrain
3    Innovation Technology & Entrepreneurship Department, College of Business & Economics, United Arab Emirates University (UAEU), Sheikh Zayed Road, Al Ain P.O. Box 15551, United Arab Emirates
*    Correspondence: nabdelwahed@kfu.edu.sa (N.A.A.A.); bwood@uaeu.ac.ae (B.P.W.)

**Abstract:** We ran two focus groups with well-educated Saudi women; there were ten participants with established businesses and ten nascent entrepreneurs. Despite the Saudi Vision 2030's centring of environmental sustainability as a key tenet of the country's development, the women entrepreneurs we studied (both established and nascent) were not well informed on the topic. Further, the well-educated women in our sample were under pressure from neither their customers nor their own religious, financial, or moral imperatives to engage in sustainable practices or seek out and implement sustainable business in any form. Our respondents believed that government should incentivise businesses to undertake sustainable practices and saw no financial benefits to initiating these practices themselves. Contrary to previous literature, we found that the women entrepreneurs in our sample did not perceive sustainability as an entrepreneurial opportunity and, in many cases, did not believe that sustainability should be an intrinsic element of any for-profit business. Our research findings imply that the prevalent top-down policy approach used by the government to promote sustainable entrepreneurial practices needs to be complimented by a more inclusive multi-actor approach that would involve local and national stakeholders. Moreover, educational policies need to promote the integration of sustainability topics within the larger educational system to promote awareness and social change.

**Keywords:** sustainability; women entrepreneurship; entrepreneurship; women; Saudi Arabia; GCC; Vision 2030





## 1. Introduction

The Brundtland report [1] established the principles of sustainable development and its related strategic imperatives; it defined sustainable development in terms of ecological, societal, and economic dimensions. Since the economy and economic development are major drivers of environmental issues, substantial efforts have been undertaken worldwide to produce and consume in a more environmentally friendly manner [2] and to align economic development and wellbeing with sustainability principles [3,4]. Since the release of the report, sustainability and environmental concerns have been added as pivotal evaluation criteria for economic development and growth [5]; these criteria take into consideration the fragilities of the biophysical environment and the finite character of natural resources [6]. In this sense, economic development must preserve ecological systems and resources "without compromising the ability of future generations to meet their own needs" [1]. Entrepreneurship, and innovation as its synonym [7], have been ascribed an active role in achieving this paradigm change and in equilibrating economic development with sustainability goals [2]. Entrepreneurship is the essence of economic development, and innovative entrepreneurial action is the greatest driver of economic growth and change [8]. Ref. [9] defined entrepreneurship as the process of opportunity

recognition, creation, and exploitation, which entails creating organisations that pursue the very same objectives [10]. In this sense, entrepreneurs are proactive and innovative, and they take risks [11] to offer solutions in the form of products and services that are aligned with sustainability principles so they can benefit individuals, communities, and societies [12].

In the Arab Gulf region, the Gulf Cooperation Countries (GCC) (Saudi Arabia, the United Arab Emirates, Oman, Bahrain, Kuwait, and Qatar) have historically mainly relied on exports of fossil fuels to form the basis of their economies. In fact, the Gulf region is one of the biggest suppliers of oil and gas worldwide, and this position has allowed the Gulf countries to generate substantial revenues and benefit from both high standards of living and high income levels [13]. Nevertheless, all Gulf countries envisage economies beyond hydrocarbons, not least because of depleting fossil fuel reserves. Countries perceive sustainability as a policy imperative of the 21st century [14], and national development plans are based on sustainability principles to help countries transition from predominantly oil-producing economies to knowledge-based economies and to move from pre- to post-industrial development [15]. Gulf nations engage fully in sustainability agendas because there is a critical need for awareness and action regarding sustainable development [16], with all countries ranking very low in terms of sustainability (Oman is highest, ranking 81st in the world).

The present paper is concerned with the Kingdom of Saudi Arabia (KSA). Perhaps even more than other Gulf states, Saudi Arabia has been extensively orientalised in English literature while also suffering from the misrepresentation and misunderstanding of its societal structure and mechanisms, particularly as these relate to women and how they live their lives [14,17,18]. An important objective of the Saudi Arabian government, shared with other countries in the Arabian Gulf, is the diversification of the economy away from oil towards other income sources, and this change in strategic focus has been the subject of research from various points of view [19–21]. The KSA has outlined its development agenda, which sets out a vision for balanced growth and sustainable socio-economic development in both its National Transformation Program of 2020 and the Vision 2030 documents [22]. The country has five broad measures for sustainability, which relate to 'personal development' (e.g., food security, health, drinking water, and sanitation), the 'environment' (referring to water and air quality), a 'well-balanced society' (referring to good governance and reductions in both unemployment and population growth), the 'sustainable use of resources', and a 'sustainable world' [22,23].

A key part of the KSA's development agenda is boosting entrepreneurship and promoting gender equality [24]. The country wants to encourage its citizenry, and women, in particular, to start their own businesses and integrate into the workforce, as well as contribute to the nation and its future [23]. With this in mind, the Saudi government embarked on a series of reforms directed at diversifying the economy and increasing entrepreneurial activity in the country. These efforts have been rewarded with the result that women and men being at parity in terms of participation in entrepreneurship, as reflected in the most recent Global Entrepreneurship Monitor (GEM) report; this is not the case in most of the rest of the world [25,26]. This regional interest in women's entrepreneurship has been reflected in the literature, with several books, research papers, and journal special issues having been produced [4,27–31]. Despite good documentation of the top-down policies and the governmental institutions facilitating entrepreneurship and sustainability in the country, little is known to date about the role of individual entrepreneurs in this endeavour. We know from previous research that an individual's values and orientations regarding sustainability are important antecedents to entrepreneurial engagement in sustainable venturing [32]. Moreover, the individual was shown to be the main agent of change in small- and medium-sized enterprises (SMEs), and individual values and beliefs are primordial in terms of the adoption of a sustainable vision and sustainable practices by entrepreneurial firms [33]. Despite being essential for the successful implementation of the sustainability agenda, individual inclination towards sustainability has not received much attention in

the literature, contrary to the top-down promotion of entrepreneurship and sustainability in the KSA and the other Gulf states.

This paper, therefore, addresses Saudi female entrepreneurs' perceptions and awareness of sustainability and the factors that influence their motivations to adopt sustainable behaviour and practices. In essence, we aim to understand the relationship between local women's values, awareness, and understanding as they relate to sustainability and entrepreneurship. We also explore challenges and possible barriers for women entrepreneurs with respect to engaging in sustainable ventures. In this context, we focus especially on green entrepreneurship, which addresses environmental and societal issues and proposes entrepreneurial solutions [12]. By gathering the responses of Saudi nationals directly, we are able to consider their lived experiences and make contributions to the literature.

This paper is organised as follows. We commence with a literature review introducing the main concepts of sustainability, entrepreneurship, women's venturing, and the relationships among these factors. We then contextualise the subject through a short description of the Saudi context. Following this, we outline the method and methodology used to provide the empirical evidence for this study. The findings are then analysed and discussed. In the conclusion, we consider our study's contribution and identify limitations and future research directions.

## 2. Literature Review

### 2.1. Sustainability and Green Entrepreneurship

The definition of sustainability was coined by the Brundtland Commission as "development that meets the need of the present without compromising the ability of future generations to meet their own needs" [1]. It refers to a development approach that encompasses and balances societal, environmental, and economic dimensions. Given the ample development challenges with regard to "people, planet, profit" [34], sustainable actions are considered vital for businesses and an important antecedent to sustainable performance [12]. Businesses and the economy bear the brunt of the responsibility when it comes to the exploitation of finite natural resources and the destruction of natural habitats and environments [35]; therefore, businesses should embrace sustainability goals to reduce and diminish their negative impact on the environment [36]. With this pivotal role, businesses also have an obligation to prevent harm to the natural environment by reducing pollution and waste [37] and by supporting environmental protective measures that involve sustainable societal/community and economic development. In fact, businesses have the capacity to impact all dimensions of sustainability with tremendous positive effects for the public good [38].

Proactive new environmental strategies [37,39] that increase environmental protection and contribute to sustainable development represent a competitive opportunity for incumbent companies [38] as well as for entrepreneurs [37]. Green entrepreneurs, or 'ecopreneurs' [40], pursue environmental sustainability goals by identifying and exploiting market opportunities by developing innovative products and service solutions [37,41] and filling market needs that were previously unserved [42]. Such entrepreneurs seek to solve environmental and societal issues by providing entrepreneurial for-profit and not-for-profit solutions [43].

Previous research has identified numerous drivers for green entrepreneurship. The institutional context plays a central role [44]; for example, numerous countries in the Arab Gulf promote sustainability as a pivotal element of their economic development and support ecologically sustainable entrepreneurship in their strategic documents (e.g., Saudi Vision, 2030). In this context, legislation and governmental regulation are primary motivators, especially for incumbent firms, to establish green sustainable practices. In their seminal article, Porter and van de Linde [38] show that investment in green technologies and processes is justified for companies because it allows them to increase their competitiveness, which is likely to serve as a great incentive. Market-driven firms have been shown to consider greener business practices in response to a sensitised consumer consciousness

regarding green issues and environmental impact [2]. Such reasons to go green(er) relate to the expectation that eco-friendliness can differentiate the concerned business from more conventional competitors and create a competitive advantage [45]. Moreover, there has been an increasing managerial awareness that environmental issues are essentially ethical and governance issues that need to be addressed proactively because they can otherwise negatively affect firm performance [46]. Companies, however, do not display homogeneous levels of sustainability: empirical research conducted in 2002 by Schick et al. [45]. allowed for a distinction between fully dedicated firms that consistently adopted eco-friendly practices, firms that were in principle open to sustainability but only partially adopted related practices, and firms that were purely compliance-driven.

Whether firms engage with green entrepreneurship also depends on individual decision makers' values and orientations linked to sustainability [32]. In 2014, Koe and Majid found significant correlations between sustainable entrepreneurship and certain socio-cultural factors, notably a long-term time orientation, a prevailing sense within a culture of the importance of maintaining harmony with nature, and social norms prevalent in collectivist contexts [47]. In fact, behavioural norms arise from cultural contexts and strongly influence sustainability orientations and values, which are typically formed during an individual's family and childhood years [45,48].

### 2.2. Women and Green Entrepreneurship

Research on gender and entrepreneurship reveals that fundamental elements of entrepreneurship are framed by gendered institutional norms, and they shape approaches to opportunity recognition and exploitation, entrepreneurial behaviour, and the general understanding of entrepreneurial legitimacy [49–51]. Worldwide, this means that, for example, men are the dominant players in entrepreneurship, and they serve as entrepreneurial role models [52,53]. On the other hand, women are "first assumed to be deficient, then 'proved' to be deficient and finally held accountable for their own deficiencies" [49,54]. This applies equally to the Arab Gulf region, where in recent years, countries have increased the convergence of female and male entrepreneurship ratios [24,55]. In addition, recent data from the Global Entrepreneurship Monitor (GEM) [25] reveal parity or near-parity in the number of female to male entrepreneurs in both the United Arab Emirates (UAE) and Saudi Arabia. Despite these encouraging developments, female venturing in the Arab region is embedded in an environment characterised by strong gender ideologies that are rooted in prevailing gendered socio-cultural norms [24]. Previous research shows how these gendered environments affect women's abilities to start and grow their ventures; for a detailed summary, please refer to the 2018 work of Bastian et al., [28] which provides a comprehensive review of women's entrepreneurship in the Middle East and North Africa and analyses the impact of gendered environments on female venturing at the micro, meso, and macro levels.

To date, there is still a scarcity of research concerned with women and green entrepreneurship or eco-entrepreneurship [56]. Yet, some research confirms that gendered notions of entrepreneurship are prevalent in practice as well as in theory [49].

In 2012, Hechavarria et al. analysed GEM data for 52 countries and found a clear gender divide regarding entrepreneurial activities that emphasised environmental or societal impact compared with traditional mainstream economic value creation. Women were much more likely to engage in the former than men [57]. U.K.-based research analysed 20 in-depth, semi-structured interviews with male and female entrepreneurs to identify differences in sustainability orientations [58]. The authors argued that women were more drawn to sustainable entrepreneurship than men because of female role models who hold strong attitudes regarding global sustainability.

These observations are in line with research findings about value-driven entrepreneurship, which was shown to be a predominantly female domain [37,59]. Hechavarria et al., in their 2012 work [57], tied this development to a hegemonic understanding and stereotyping of entrepreneurship as being male and focused on pecuniary value creation. In

that sense, female social and green entrepreneurs are perpetuating the hegemonic notion of entrepreneurship by sticking to their culturally assigned roles [57]. A recent study by Xie and Wu [60] confirmed this: it showed how responsible entrepreneurship actions had a greater impact on the venture success of female entrepreneurs than that of their male counterparts; the authors explained this fact with respect to gendered expectations regarding the venture performance of women and men. Previous research corresponds to eco-feminist approaches rooted in essentialism, which argue that women, by virtue of their biology and their naturally assigned roles as child bearers and caretakers, have a closer relationship with nature that makes them more likely to engage on behalf of nature [61].

The notion, however, that women have an exclusively female approach or a female essence that determines their green and eco-friendly behaviour has been refuted both by social science [62] and by eco-feminism approaches applying a social constructivist perspective that put forward a more convincing explanation linking the female predisposition for green and eco-friendly issues to women's ranks in societies, where they tend to fill caregiver positions, hold part-time and underpaid jobs, and often work in low-status sectors and domains. Consequently, in most societies, there are still significant gender gaps when it comes to salaries, access to diverse resources, access to labour markets, and poverty; these factors render women more vulnerable with regard to pollution and environmental hazards [61], for example, the effects of climate change [63]. The literature on environmental justice shows that women are more likely to be attentive to sustainability topics since they are socio-economically more vulnerable and, worldwide, they tend to be disproportionally negatively affected and disadvantaged by natural disasters (e.g., in 1991, a cyclone in Bangladesh killed almost 140,000 people, of which 90% were women and children) [64]. Women experience more of the negative effects of environmental hazards and develop strong beliefs about the harmful consequences of pollution, climate change, and the like [65], which explains the greater likelihood of women engaging in environmental action. Women are also more socialised for altruism than men, which leads them to pay attention to the needs of others and to support and help their communities [66]. Individuals with a strong tendency to help tend to also be more aware of the negative impacts of their own behaviours on others and on the environment as a whole; they tend to feel more personally responsible [59,66,67].

### 2.3. Drivers of Sustainable Entrepreneurial Action

Previous research has identified diverse drivers behind sustainable and green entrepreneurship, such as perceived entrepreneurial desirability, i.e., the general attractiveness of becoming an entrepreneur [3,67–69]; perceived feasibility and self-efficacy [3,70]; intrinsic and extrinsic motivations [3,67,71]; and individual attitudes towards the role of sustainable development and the preservation of nature [72]. In fact, 'attitude' regarding sustainability and green values [73] was shown to be the strongest element in terms of pulling individuals into green business [3,72,74]. Moreover, altruism [75], the desire to create social value [75], and social orientation [76] have been shown to play pivotal roles in the decisions of individuals to engage in sustainable or green business [3,77]. The normative fundamentals of female entrepreneurship have been shown to be strongly influenced by social and sustainable values [76]. For example, men tend to engage in green entrepreneurship when it represents a business opportunity, whereas women engage in green entrepreneurship for the cause itself, and they want to create environmentally friendly and inclusive services and products [78].

### 2.4. The Saudi Context

The Gulf Corporation Council (GCC) area consists of six countries (Bahrain, Kuwait, Oman, Qatar, Saudi Arabia, and the United Arab Emirates); they are all Arab and Muslim but cannot be fairly represented as a monolith, either religiously or culturally, although this is often the way the region and its beliefs are portrayed [79,80]. The Kingdom of Saudi Arabia is most widely known as a petro-rich state at the heart of the Middle East and the

Gulf. With a GDP of USD 700 billion, it is understood as one of the richest countries in the world. Saudi Arabia is a pivotal location for the world's 1.8 billion Muslims [81] as it is the birthplace of the Prophet of Islam, Mohammed. It is also the location where much foundational Muslim history took place and hosts the two most important mosques and sites of pilgrimage in the western part of the country: Al Haram in Makkah, and Masjid Al Nabawi in Medina. Saudi has a resident population of 35 million people, of whom 71% are aged between 15 and 64 [82] (accessed on 20 May 2022)) and 93% are Muslim [83]. Additionally, 35% of the total population consists of (mostly male) temporary migrant workers, whose numbers inflate the overall male population to 73% of total residents. The climate of the country is mostly desert with very low rainfall—3 inches a year in the eastern provinces, which is where our data were collected—but even within this general characterisation, there is a lot of topographical variety, including mountainous areas and areas to the north in which snow falls each winter.

In Saudi Arabia, the concept of entrepreneurship is deeply rooted, in particular, with respect to commercial businesses, with a history of travelling caravans of goods crossing the land for millennia and the location of important pilgrimage sites in the country amplifying the experience and extensive involvement of its people in business and trade. More recently, the Kingdom has played a central role in the energy market and has been developing policies to promote growth in the private sector and simultaneously achieve sustainability. To this end, the government has launched Vision 2030 projects with the aims of prosperity, sustainability, and development. Notably, sustainability has been one of the key pillars of the Saudi 2030 vision. The Kingdom has launched several ambitious initiatives, such as zero carbon projects, renewable energy technology, environmental preservation and protection, and liveable green cites (for example, 'The Line' project). In these ways, the Kingdom of Saudi Arabia is taking essential steps and pursuing reforms in order to realise a greener, cleaner, and more sustainable country [22].

However, the legislation and by-law reforms concerning sustainability are facing challenges with respect to cultural acceptance [84]; as such, success in linking the reforms to the ways in which communities are going to benefit is believed to be the best approach to resolving many of these issues, and once the benefits are clearly understood, objections will be quelled.

## 3. Methodology

### 3.1. Qualitative Methodology

We adopted a qualitative approach to investigating Saudi female entrepreneurs' awareness and adoption of sustainability behaviours and practices, as well as the factors that influence them. A qualitative method provides flexibility in interactions with study participants and allows a holistic view, the collection of rich data, and a deep understanding of the perspectives that respondents have regarding their experiences and the practices of sustainability activities carried out within the Saudi context. As our respondents were all female, feminist research approaches were adopted for this study, in keeping with recommendations from leading authors in the field [85–88], in order to better understand their activities and experiences associated with entrepreneurship and development [89,90]. This 'conversational' approach—and, indeed, the focus group model—is a research approach naturally supported by cultural and religious norms in the Gulf [17,91].

In order to develop a deeper understanding of Saudi female perceptions, understanding, and commitment with respect to sustainability, the study targeted Saudi women of two types: current, experienced entrepreneurs and nascent entrepreneurs with the idea of launching a new business. The study was conducted in the eastern governorate (Al-Sharqiah) of Saudi Arabia. The eastern province encompasses the entire east coast of Saudi Arabia and acts as a major platform for most of the Kingdom's oil production and exports.

*3.2. Sample Selection and Method*

Data were collected using focus group discussions with 20 Saudi women purposefully selected using criterion sampling and divided into two groups. The first group included 10 Saudi female MBA students who expressed an intention to start their own business and had attended an entrepreneurship course. The second group included 10 Saudi women who already owned and had been running businesses for more than three years (as per the GEM parameters for established businesses) from a range of fields and across the service and industrial sectors.

Prior to the focus groups taking place, a focus group discussion protocol and general topic themes were prepared as a guide. Before starting each session, there was a discussion of the confidentiality of participants' data and the privacy of disclosures; participants were assured of anonymity. The focus group itself was conducted and facilitated by the authors and was audio recorded; in addition, field notes and journal entries were collected. Interviews were conducted in English as all the participants were bilingual, but translations into Arabic were provided by one of the authors whenever necessary. The recordings were transcribed verbatim. Each group discussion lasted 60–90 min. During the focus group discussions, researcher objectivity and biases were carefully monitored. Both qualitative and observational data were analysed using a thematic analysis approach over consistent coding and theme generation to represent the Saudi female entrepreneurs' perceptions and experiences.

*3.3. Data Analysis*

An inductive and interpretative approach using thematic analysis was adopted to analyse the various data types. The themes and patterns that emerged from the data and the analysis thereof were drawn from all the data collected from the focus groups' discussions, the field notes, and the authors' observations. The theme generation process drew on the five well-established phases of data analysis developed by [92]. These phases include familiarisation with the data, the identification of codes and patterns, the linking of codes that relate, and the development of categories that incorporated the themes that emerged. This method, which involved the manual reading and re-reading of the transcripts, was conducted by the authors to identify themes and develop codes [93]. From this iterative process, five key themes emerged, as detailed below (Sections 4.1–4.5).

## 4. Findings

The participants of the two focus groups were located in the cities Al-Khobar, Dammam, Qatif, Hufuf, and Jubail in the eastern province of Saudi Arabia. The participants had different types of businesses, including coffee shops, an art gallery, a fuel and petrol trading business, an HRM development and training business, a clothing business, a chocolate and gifts business, a spa and women's salon, retailing companies, and a hospital. Among the invited women entrepreneurs, three were initially working for their family businesses and had started their own small firms at the time the focus groups were conducted. Eight of the participating entrepreneurs were running registered businesses, while two were running unregistered businesses.

A thematic analysis of the transcripts, notes, and observations relating to the focus groups resulted in five main themes that are detailed in the sections below.

*4.1. Perceptions and Understanding of Sustainability*

In response to a general question on the meaning of sustainability and its main domains and objectives, it was noted that there was a general lack of awareness of sustainability, its definition, and its parameters. Even though all the participants in both groups were highly educated and all of them had already obtained college degrees, they had very unclear views about the meaning of sustainability and its goals. The female participants of both focus groups lacked knowledge of sustainability, and either their answers were that they 'do not know exactly' or they gave an inaccurate or incomplete answer.

The majority of nascent female entrepreneurs who had attended an entrepreneurship course did not have any idea about the meaning of sustainability; some answered: "No, I have no idea about it, I just heard general information about it in the news" (Participant N. 1, group-1-MBA female student). Another female nascent entrepreneur added: "sustainability is related to the environment" (Participant N. 5, group-nascent female entrepreneur).

In addition, while the female entrepreneurs expressed that they had acquired some information, they still had blurred ideas about the meaning of sustainability: "Yes, we heard about it, but we do not have enough information about it. We know that it's one of the country's initiatives from its Vision 2030 and also, I heard it's a trend in the EU. I also heard about it during the B20 and G20." (Participant N. 2, group-2-entrepreneurs).

Despite the initial lack of information and awareness, all the participants—once having had a definition of sustainability supplied to them—agreed on its importance, i.e., that it was crucial for ensuring quality lives for both current and future generations. The nascent entrepreneurs' group was able to give a clearer answer as to why sustainability is important: "I think it is important to improve the quality of our lives. And the maintenance of the resources for future generations" (Participant N. 4, group-1-nascent entrepreneur).

Established female entrepreneurs, i.e., those in the second group, seemed more assertive about the importance of sustainability. This may be because of their entrepreneurial experience acquired through activities and interactions while running their enterprises. One answered in detail:

> "I think sustainability is a critical issue for youth and the current generation, especially business owners. I believe it is important to spread information about it since it is critical for economic, intellectual and even academic development. In terms of how to implement the academic and theoretical research in all aspects of sustainability and at all levels, not just companies (are involved). This is my point of view, there are so many individuals who would like to participate in sustainability, however they do not know how to or which direction they should follow. They think it's quite a difficult thing to do, due to a lack of understanding, a lack of tools, or an inability to use them" (Participant N. 8, group-2-entrepreneurs).

Another entrepreneur added: "It is important and this is because of the increase of unemployment rates and climate change which threatens the future. There are so many studies that claim in the future there will be more climate change and increases in poverty rates, of course because all of this is important. We need to plan ahead for those environmental issues before a catastrophe happens" (Participant N. 10, group-2-entrepreneurs).

Obviously, current female entrepreneurs have more potential to participate in sustainability-related activities and incorporate them into their entrepreneurial activities than young women working in start-ups or nascent entrepreneurs. However, they did not take action, perhaps because they lacked both knowledge and motivation.

### 4.2. Sustainability Is Not an Opportunity

Saudi women in both groups did not seem enthusiastic about adopting sustainability or incorporating it into their business activities. When discussing the possibility of incorporating sustainability-related activities into their businesses, they focused on recycling and looking for new materials and organic ingredients. One entrepreneur explained that business was not emotional and a focus on productivity was dominant unless they had corporate responsibility initiatives in place: "In terms of business there is no emotion; their decisions are mostly rational, so if there is no reward or incentives they will not act unless the company has a specific commitment towards social responsibility. In this case sustainability will be part of their actions".

Another participant articulated that she did not think of incorporating sustainability into her business activities: "I believe each business has its nature and characteristics that could have different sustainable implications. Since I am working in the field of electronic marketing and trade, I did not in fact take any actions or practice sustainability yet. But I am

thinking about this issue currently and maybe this conversation will stimulate my thinking and help me to figure out how I could participate in sustainability activities through my business" (Participant N. 7, group-2-entrepreneurs).

One entrepreneur stated that the discussion drew her attention to sustainability issues and said that she would take action regarding recycling in her business: "I have two business, my aim is to establish my own brand in the business. So, my aim is to ensure continuity and survival for my business. Thus, I will try to educate myself about sustainability since I don't have much information or detail about it. At that point, I can take action in terms of preserving resources and reducing waste, recycling" (Participant N. 6, group-2-entrepreneurs).

Another respondent, who had a hospital enterprise as a family business, mentioned that they tried to practice sustainability. However, all their efforts had been unrequited due to infrastructure problems:

> "I guess of course it's possible, but if the level of awareness has been raised and there is a missing link, yeah, the recycling process has something missing . . . like in our hospital we had an experience where we had a recycling program for paper. We ended up with stacks of paper because we don't have a recycling paper factory in the Eastern region, so there was a missing link. Yeah, the cycle was not complete. I mean it's not a one man show, it's for everyone, and everyone has their own task" (Participant N. 8, group-2-entrepreneurs).

Furthermore, a participant indicated that she had become involved in sustainability by accident, as she took action to reduce a business cost, and it turned out to be a sustainable act as well:

> "In my chocolate business, for example, I started a recycling initiative. I offered the customers who return the chocolate packages to refill it for a discount to encourage them to recycle the packages and reduce the waste and harmful packaging . . . . . . . . . For me the practice in fact was at the beginning of my business to reduce my costs. However, the act continued later on as participation in protecting the environment, and it turned out to be a sustainable act and became one of my business activities and responsibilities not just with an aim to cut costs" (Participant N.5, group-2-entrepreneurs).

However, nascent entrepreneurs indicated that they believed sustainable businesses added more value to the economy and the community: "I think being a sustainable enterprise, any of it, will give you value, and also the community" (Participant N. 4, group-1-nascent entrepreneur).

It is noted that despite the belief in the importance of sustainability amongst Saudi female entrepreneurs and its obvious value, they did not see sustainability as a business opportunity for them.

*4.3. It Is a "Top-Down" and Not a "Bottom-Up" Initiative*

In response to a general question about how entrepreneurs can participate in sustainability, the participants' responses centred around the belief that entrepreneurs can tackle actions that relate to the employment rate among Saudis (through Saudisation, i.e., the policy of requiring businesses to have a certain percentage of Saudi nationals on staff), resources, and renewable energy. However, they expressed the opinion that these efforts should be initiated, enforced, and regulated by the government, otherwise not much can be achieved since businesses' main aims are wealth generation and profitability. From their point of view, it is not an individual but a governmental, task:

> "I suggest that there should be restrictions and governmental regulations to force sustainability. For example, people stopped throwing rubbish in the street when the government enacted fines for the violators" (Participant N.9, group-2-entrepreneurs).

Participants also believed that sustainability could be achieved if organisations adopted a sustainability culture and raised awareness, which they said should be guided by academic research and governmental efforts: "I think if all the organisations—governmental and nongovernmental—took the initiative to adopt sustainability culture and raise sustainability awareness, it could be done via governmental and academic research and guiding individuals' practices" (Participant N.5, group-2-entrepreneurs).

Another participant said: "I would like to stress the role of governmental regulation; without regulation it will be difficult to achieve the government's aims. I am saying that the government should directly issue fines and sanctions; but it should start by spreading its vision, raising awareness, educating, and inviting people and businesses to get involved and act, then there should be strict regulations for not following or violating these regulations (Participant N.2, group-2-entrepreneurs).

With respect to the nascent female entrepreneurs' point of view, they also believed that government could push this agenda better than individuals or enterprises: "The government is doing its best to solve most of the problems and takes on many initiatives. For example, at the university we have two initiatives for the 2030 vision: one is related to good agriculture practices and the second is the (Saudi) employment gap. Currently, we also have the green Saudi Arabia initiative which is initiated by Prince Mohammed bin Salman. And also, at our university, King Faisal University, they participate in this big project" (Participant N. 8, group-1-nascent entrepreneur).

Furthermore, study participants suggested that the government's role was to motivate businesses by incentivising and rewarding sustainable activities: "The sustainability goals could be achieved through encouraging the community to recycle, for example when I was in the U.S. I saw the society trying to maintain cleanness through providing garbage boxes for soda and water cans and there was a machine in which we put the sodas cans, so the machine gives you a bill, which you give to the cashier, and he gives you back money as an incentive for your recycling. This can really encourage the community" (Participant N. 2, group-1-nascent entrepreneur).

### 4.4. Socio-Cultural Structure Is Reflected in Passive Behaviour

Study participants pointed out that there was a relationship between adopting sustainability and the Saudi socio-cultural structure. Saudi culture and values are defined by Islamic heritage and Bedouin traditions. Saudi people are characterised by their strong moral codes and values, such as loyalty and a sense of obligation to support their community. One participant added that she did not believe that the current Saudi culture was in favour of sustainability and did not support its practices:

"Yes, as a Muslim community we should be committed to preserve resources and save/preserve grace (Ar: *hifz alniema*) and not be extravagant with resources, all of these issues are sustainability practices. However, I believe up till now the social culture of our society as Saudi people does not support sustainability in general, as individuals or enterprises. Despite what (another participant) mentioned where our Islamic religion asserts and encourages sustainability behaviour, in fact on the contrary, we use everything extravagantly. Let me give you an example, one of our slogans at the university is to be a "paperless University". However, all our work is paper based. So, we suffer from a shortage of paper and printer ink, because we extensively use them in all communications and transactions using traditional ways" (Participant N.4, group-1-nascent entrepreneur).

One participant suggested that Saudi people would act to achieve sustainability goals if they recognised and realised the importance of sustainability and its impact on their community:

"In terms of socio-cultural aspects, the majority of people don't have any idea about sustainability and generally lack awareness. Saudi people, in addition to being true Muslims, are so patriotic and love their country, they get so emotional

towards anything that would support their country. So, if we relate sustainability to religion and patriotism, they would do anything and I believe they will act more to achieve sustainability goals" (Participant N.3, group-2-entrepreneurs).

It is noted that all participants reported that despite culture and values strongly rooted in Islamic beliefs, the practices of Saudi people and entrepreneurs did not support the achievement of sustainability goals. Thus, there is a need to raise awareness and spread information about it, in addition to giving examples of best practices.

*4.5. Challenges for Sustainability Involvement and Participation*

Participants in both focus groups reported several obstacles and challenges for Saudi entrepreneurs seeking to practice, incorporate, and get involved in sustainability activities. Those challenges included a lack of awareness, a lack of governmental regulation, a lack of motivation, a deficiency of incentives and rewards, and the nature of the country and its climate.

One participant explained that a lack of motivation was one of the challenges for sustainability: "In fact it depends on the motivation of the whole community. When the whole community is motivated, the individuals will be motivated too" (Participant N.2, group-1-nascent entrepreneur).

In addition, another participant referred to the lack of tools and technology and the requirement for collaboration between the government and entrepreneurs as one of the obstacles: "From my point of view the main barrier is the difficulty of executing some of the sustainability aspects, and lack of tools. In addition to this, the barriers created by a lack of cooperation and collaboration between countries, governments and business, because some of sustainability solutions or actions require a high level of intense technology which cannot be achieved without partnership" (Participant N.3, group-2-entrepreneurs).

In addition, all of the participants suggested that Saudi entrepreneurs would take serious steps towards the achievement of sustainability only if there were governmental regulations, enforcement, and motivation:

> "From my point of view, I do not think there is anything different about Saudi people to drive them towards sustainability. They are just like anyone else around the world. On the contrary, up till now we do not have sufficient awareness about sustainability and I do not believe we, as Saudi people, will take serious actions towards sustainability without the help of the authorities and strict legislation for both people and organisations. There should be motivation provided for them. If there is no such motivation, they will not get involved. There are already many regulations for sustainability which have been in place for a long time already. Take, for example, when the government required planting outside the populated cities, to reduce air pollution in the cities and protect people's health. Then we started to notice specific terms for plants and any type of activities that would cause harm to the environment" (Participant N.3, group-1-nascent entrepreneur).

Furthermore, one of the participants stated that sustainability in Saudi Arabia would always be challenged by business practices that create pollution and global warming:

> "There are so many (construction) developments in Saudi Arabia and there are many plans running currently and there will be always plans. However, business practices such as pollution and global warming, will always constitute a challenge for us and the country to overcome" (Participant N.7, group-2-entrepreneurs).

In addition, a participant pointed to the high growth rate of the population in Saudi Arabia as constituting a challenge due to the increased consumption of resources: "Maybe because the population growth in Saudi Arabia is quite high, there is an increase in consumption rates. When the consumption rates increase then there will be more waste, which creates difficulties for sustainability and more damage to the environment" (Participant N.4, group-1-nascent entrepreneur).

In terms of nature and climate, the majority of the Saudi Arabian land area is characterised by a desert climate, except in the southwestern part of the country, which exhibits a semi-arid climate. The summers in the Saudi central region are extremely hot and dry on a regular basis. Thus, Saudi people are not able to sense the role of sustainability and the effect it will have on the nature of the county except perhaps in terms of slightly cleaner cities.

> "I think this should be a concern, for all of us and each one of us should have his own task towards sustainable water in Saudi Arabia because we don't have enough water in Saudi Arabia and we have a very dry climate. We have to realise that pollution is growing day after day. So, I think, each person should have his own task. We have limited water and natural resources in Saudi Arabia" (Participant N.2, group-2-entrepreneurs).

## 5. Discussion

Despite the fact that much of the literature around sustainability and its particular relevance to women assumes that women are both generally (socialised to be) more caring [58,59,66,67] and more likely to suffer from the effects of environmental injustice [61,63,64], our sample reflected neither of these perspectives.

Saudi Arabia is a large country with varied eco-scapes, and the women we interviewed were from the eastern provinces where there is sea/coast; however, the country is mostly a desert landscape. One might assume that the harshness of the natural environment would motivate environmental engagement and action [61], but instead it appears to engender disassociation. This may be because climate change is not obvious in this already very stark environment, and it may also point to the use of public spaces being different to what is observed in many other places. For example, public parks and the like are mostly used at night, as the daytime temperatures for much of the year are prohibitive for sitting around outside (this makes the appearance of parks less noticeable). In addition, outside public spaces are more often filled with men than women, and all our respondents were women [17,94,95].

Moreover, the Saudi people traditionally lived as Bedouin and are, thus, used to moving across and surviving in a very harsh landscape. Since their country became a petro-state, Saudis have mostly left the desert and pursued lives of comfort in the cities, replete with the benefits of modern technology that great wealth can bring. As such, there is a perceptible gap between the abilities, priorities, and mentalities across generations [91]. "Post-oil" Saudis have a very paternalistic view of government; the country used its national wealth to provide healthcare and allowances for Saudi nationals to boost their standard of living [14]. As such, Saudi nationals look to the government for care and reward, hence the views of our interviewees that the government should incentivise their participation in sustainability initiatives if they expect buy-in.

Further, as a 'developing' country, Saudi has a short but very active history of infrastructure development. At present, sustainability is part of the national strategy and vision documents, but investment in the equipment required for safe and sustainable recycling, for example, has not yet occurred to an extent sufficient to meet the relevant goals. Our respondents again looked to the government to fill this gap before undertaking any resolution to prioritise sustainability in their own current or future businesses. They saw no benefits in pursuing it—independently or otherwise. Instead, they understood sustainability as a business cost rather than stewardship for future generations (as one might expect from a Muslim population) or a sensible business strategy ensuring the longevity of their investments [38,45].

Our respondents also noted that there was no demand for sustainable practices from their customers, further reflecting a lack of awareness, a feeling of irrelevance, and disconnection from the natural environment. Remarkably, the one respondent who did engage in a sustainable practice (refills) did so by accident. Overall, the views of our respondent entrepreneurs reflected their status in a consumption-based society, i.e., they looked to buy

their way out of bad situations and took a passive and materialistic approach to the issue of sustainability and their related responsibilities as business owners and individual citizens.

Our established entrepreneurs were less aware of sustainability issues and worried about cost versus profit in terms of doing more to support sustainability; that is, cost was seen as a barrier to more engagement. Our nascent entrepreneurs had higher levels of awareness but also did less, as they had less experience.

Overall, our respondents were completely inconsistent with what might have been expected from female entrepreneurs in general and across the rest of the world, as reflected in the canon. Their collective cultural, religious, and gender affiliations did not manifest in the way the literature led us to expect, and our respondents instead reflected a range of positions, with regard to the environment and the government's message, that was characterised by a lack of awareness and a lack of proactive engagement. Several issues may have contributed to these positions. One is the 'normative male' model of business and entrepreneurship sacralised in the literature as the universal ideal. In this case, with a pecuniary focus at the forefront of their thinking, our respondents were more aligned with male views, as per the research of Hechavarria et al. conducted in 2012. Another potential issue is that the respondents' culture is disconnected from nature because the environment is already quite stark; recent cultural changes, according to which many have changed their way of life by moving to cities and being separated from a natural landscape that does not easily show signs of climate change, have exacerbated this situation. In addition, since the discovery of oil and its associated enormous wealth, citizens have come to rely on their government for care and support; concomitantly, they trust the government and look to it for incentives to buy-in to its sustainability vision.

As the sustainability focus is relatively new, there is no well-established infrastructure in place for the mechanistic delivery of easy participation for those willing to co-create these governmental priorities. The fact that the infrastructure is not in place means that more motivation needs to come from the businesses themselves—and, with that, coverage of the extra costs—for which there is little incentive as there is no demand for this from customers of the businesses. Saudi is most recently a consumption-based society rather than a production-based one, and this further exacerbates the phenomenon of all the citizenry looking to the top/government to take the lead.

Another possible reason for the limited interest in and uptake of sustainability issues compared to what sustainability literature might have predicted may relate to the relationship between the Kingdom and the literature. Saudi Arabia has long been largely closed to foreign visitors, with the first tourist visa system being created in 2019. As such, visitors were, until recently, only accepted either for religious pilgrimages or business and faced clear restrictions on where in the country they could go. As such, despite 30% of the population being expat workers, Saudi nationals would intuitively be expected to reference local over global concerns. Although global attention has had a sustained focus on climate change, a delay or suppression of interest and uptake by the Saudi citizenry might be expected, especially with the 'change' part of climate change being a lot less obvious in this part of the world.

With the increase in research publications on Muslim consumption in general [96–99], as well as the increase in publications both on the region and on women's entrepreneurship in the region, important insights into why (and how) respondents may be quite divergent from expectations engendered by the wider literature can be both studied and uncovered. Such an investigation represents an important contribution to the literature and is something that has been repeatedly called for by leading scholars [14,18,100] in order to "spotlight conditions" [101] and develop informed research designs [17,102]. This study contributes to the literature in all of these ways.

## 6. Conclusions

Our research analysed the perceptions and awareness of female entrepreneurs in Saudi Arabia regarding sustainability and the entrepreneurial opportunities related to it. We

also wanted to identify the factors that impact individual motivations to adopt sustainable practices and sustainability as part of the entrepreneurial business model. Major findings include the incomplete and insufficient knowledge and understanding of sustainability issues exhibited by individual entrepreneurs. Female entrepreneurs in our sample did not perceive sustainability as an entrepreneurial opportunity and, in many cases, they did not consider sustainability as something that should be an intrinsic element of any for-profit business. Moreover, individuals showed a lot of reliance and dependence on the government to initiate sustainability action on all levels. In fact, sustainable practices were not considered voluntarily unless there were clear prescriptions and regulations promulgated by the state that obliged individuals to engage in sustainable actions. A main explanatory variable that was offered to aid in understanding these results was the socio-cultural structure that promotes more passive behaviour by individuals when it comes to sustainability. Our findings are a bit sobering with respect to the ambitious sustainable development goals of Saudi Arabia as reflected in their economic vision for 2030. One main problem may be that the current sustainability message in Saudi Arabia is top-down, i.e., it flows from the government in the form of strategic plans, and it is slow, as many public marketing campaigns are, to trickle down. At present, the message is not being actively taken up by either individual members of the public or the entrepreneurial classes. Recent research about women's entrepreneurship initiatives [103] shows that sustainability implementation and performance are closely tied to the organisation and governance of such initiatives. Instead of top-down prescription, policy makers should involve multiple actors on the local, regional, and national levels to promote sustainability and develop related initiatives in connection with entrepreneurs. A multi-actor approach would enable the necessary social change based on shared ideas and understanding regarding sustainability, and it would create a broad social acceptance and legitimacy regarding sustainable practices as an intrinsic part of any business. Saudi Arabia has some experience involving, for example, women's organisations, such as the Women's Section of the Chamber of Commerce in Jeddah and Riyadh, in women's development planning [24]. The government needs to be more proactive; it needs to involve and include local and sectional women's organisations and women's social movements to engender broad behavioural and attitudinal changes about sustainability and business. Moreover, educational policy needs to be aligned with sustainability goals and promote teaching about sustainability as an intrinsic element within the larger educational system in Saudi Arabia; such policies could link local and national concerns to global social, environmental, and economic issues, as well as teaching values-based and multi-disciplinary sustainability content in schools and universities. Such educational reforms can engender social change through individual awareness and engagement with sustainable development. It has been observed by authors examining cultural and tourism practices in the Gulf that local culture allows inclusionary discourse to enact exclusionary practices [104]. This process may potentially be harnessed by the government to promote sustainability practices by individuals and businesses, allowing top-down efforts to be met by bottom-up engagement.

Overall, our paper challenged both the 'inherent female interest' and 'primarily affected population' views in the literature of women entrepreneurs and their engagement with green business, with our respondents reflecting a more 'male', pecuniary viewpoint on engaging with sustainability, i.e., if the government did not incentivise sustainability, they saw no benefits to them as for-profit entrepreneurs.

Further research into sustainability with Saudi women entrepreneurs could look into the many unsupported assumptions contained in the literature in other contexts in order to examine why these elements are not present, as well as how to best engage these entrepreneurs to encourage co-creation and partnership with government to realise developmental objectives and how to increase awareness and consideration for the environment and climate change in general [105]. The nexus of sustainability and entrepreneurship is intertwined with existing cultural and institutional contexts that present challenges for

successful sustainability implementation. Future research and entrepreneurial critiques need to be open to more diverse cultural and institutional perspectives.

**Author Contributions:** Conceptualisation, N.A.A.A. and B.L.B.; methodology, N.A.A.A., B.L.B. and B.P.W.; formal analysis, N.A.A.A., B.L.B. and B.P.W.; data curation, N.A.A.A.; writing—original draft preparation, N.A.A.A., B.L.B. and B.P.W.; writing—review and editing, B.L.B. and B.P.W.; funding acquisition, N.A.A.A. All authors have read and agreed to the published version of the manuscript.

**Funding:** This research was funded by King Faisal University Grant #425 and mentored through the MENAGEN (Middle East and North Africa Gender Entrepreneurship Network) Mentoring for Publication Programme 2021 (MPP), sponsored by the Babson Global Center for Entrepreneurial Leadership.

**Data Availability Statement:** Not applicable.

**Conflicts of Interest:** The authors declare no conflict of interest.

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
