# Peer review of "Women, Entrepreneurship, and Sustainability: The Case of Saudi Arabia"

_sustainability, doi:10.3390/su141811314_

Round 1
Reviewer 1 Report
Firstly I would like to thank the authors for the opportunity of reading and comment on their work. The topic raised is very interesting and still overlooked in the literature. The entrepreneurial endeavors in Islamic countries present idiosyncracies deserving further analysis.
The connection between entrepreneurship and sustainability with gender is very interesting. In my opinion, the literature review needs some additional aspects, mainly in regard to gender issues. In this vein, I suggest some recent works in gender for these geographies:
Costa, J. and Pita, M. (2020), "Appraising entrepreneurship in Qatar under a gender perspective", International Journal of Gender and Entrepreneurship, Vol. 12 No. 3, pp. 233-251. https://doi.org/10.1108/IJGE-10-2019-0146
Costa, J. and Pita, M. (2021), "Entrepreneurial initiative in Islamic economics – the role of gender. A multi-country analysis", Journal of Islamic Accounting and Business Research, Vol. 12 No. 6, pp. 793-813. https://doi.org/10.1108/JIABR-01-2020-0010
As such, the literature review needs to be further developed by introducing the importance of gender in these geographies due to the idiosyncrasies of these countries.
Also, section 3 needs deep re-structuring as the main aspects of the empirical experiment are not clarified to the reader.
The results section does not compare the present findings in extant literature. There is a need to establish a direct comparison between what is found here and what was previously found.
The conclusions need to be stronger and the contributions of the study need to be clear in both the theoretical and the empirical domains.
Please add a recommended policy package based on the present findings as the most relevant aspect in the research in entrepreneurship is what should be done to improve the ecosystem.
Best of luck in your research!
Author Response
Reviewer 1
|
1 |
Extensive editing of English language and style required
Response: We have sent the article to a professional copy editor and hope that the result fulfills your expectations. |
|
2 |
Firstly I would like to thank the authors for the opportunity of reading and comment on their work. The topic raised is very interesting and still overlooked in the literature. The entrepreneurial endeavors in Islamic countries present idiosyncracies deserving further analysis
Response: Thank you very much for this encouragement; as authors we are very engaged in these topics. If you are interested in gender and entrepreneurship in the MENA region, we suggest for you to connect with MENAGEN (a research network of likeminded researchers and very supportive) – Dr. Haya Al Dajani would be a good contact – please, note, she is not an author of this paper.) |
|
3 |
The connection between entrepreneurship and sustainability with gender is very interesting. In my opinion, the literature review needs some additional aspects, mainly in regard to gender issues. In this vein, I suggest some recent works in gender for these geographies: Costa, J. and Pita, M. (2020), "Appraising entrepreneurship in Qatar under a gender perspective", International Journal of Gender and Entrepreneurship, Vol. 12 No. 3, pp. 233-251. https://doi.org/10.1108/IJGE-10-2019-0146 Costa, J. and Pita, M. (2021), "Entrepreneurial initiative in Islamic economics – the role of gender. A multi-country analysis", Journal of Islamic Accounting and Business Research, Vol. 12 No. 6, pp. 793-813. https://doi.org/10.1108/JIABR-01-2020-0010 Response: We cite the first reference as requested in our paper to provide further evidence for the increased interest in the subject of entrepreneurship and gender in the MENA region. |
|
4 |
As such, the literature review needs to be further developed by introducing the importance of gender in these geographies due to the idiosyncrasies of these countries. Response: Thank you for this proposition: the literature review contains a whole section dedicated to “women and green entrepreneurship” – within this section, we address gendered entrepreneurial norms, gendered entrepreneurial theory and gendered practices and we make it very clear that introducing gender is fundamental to ALL contexts (please, note that the female/male ratios are substantially better in the Arab Gulf than in European, North American or other Western contexts – see the concerned GEM reports). We, however, focus on literature concerned with women and green entrepreneurship and critically evaluate it instead of detailing at length the gender debate in entrepreneurship research. Yet, we have integrated a paragraph pointing to the situation of the Arab context. |
|
5 |
Also, section 3 needs re-structuring as the main aspects of the empirical experiment are not clarified to the reader. Response: Section 3 has been extensively restructured for clarity of our process and methods. Several references relevant to our approach have been added. |
|
6 |
The results section does not compare the present findings in extant literature. There is a need to establish a direct comparison between what is found here and what was previously found. Response: Thank you for this proposition. We have, however, refrained from mixing the findings with previous literature. Typically, this should be done in the discussion section, where we in fact address the findings and put them into perspective with regards to existent literature. We tie back findings of our literature review, and clearly show that our results capture little of what has been discussed in previous literature. We show how the women’s responses in our sample regarding sustainability awareness and sustainability engagement reflect a position associated with predominantly male perspectives (rather than women caring about environment and community). In response to your comment, we have made sure to include more literature and citations in the discussion part.
|
|
7 |
The conclusions need to be stronger, and the contributions of the study need to be clear in both the theoretical and the empirical domains. Response: We have strengthened the paper by adding more to our discussion section and drawing through those points to highlight our contributions and also to make suggestions on how the issues we raise might be addressed for a better uptake and participatory response to sustainability initiatives in the Kingdom. |
|
8 |
Please add a recommended policy package based on the present findings as the most relevant aspect in the research in entrepreneurship is what should be done to improve the ecosystem. Response: Thank you very much for this comment. We have strengthened our conclusion part and propose concrete policy recommendations referring a) to a more inclusive multi actor governance approach that complements the top down policy approach, as well as b) educational policies that include sustainability as intrinsic part into school and higher education curricula |
Reviewer 2 Report
This is an interesting paper and I enjoyed reading it. However, there are essential weaknesses that need to be addressed.
0) Abstract: Authors should state their contribution in terms of issue problems solved or ameliorated, theory or policy dilemmas resolved, or the like. Abstract should offer at least one example of a theoretical or managerial implication that authors concluded after their work.
1) The introductory/opening section should communicate a little clearer the literature gaps, as well as the study's aims & objectives in order to facilitate the flow of the study.
2) Additional references to recent & relevant empirical studies could increase the quality of the research paper and provide a much clearer message to the reader - these may help you building your discussion which needs to be extended. Add the following to your reference list:
Andati, P., Majiwa, E., Ngigi, M., Mbeche, R., & Ateka, J. (2022). Determinants of Adoption of Climate Smart Agricultural Technologies among Potato Farmers in Kenya: Does entrepreneurial orientation play a role?. Sustainable Technology and Entrepreneurship, 100017.
Dijkstra, H., van Beukering, P., & Brouwer, R. (2022). Marine plastic entrepreneurship; Exploring drivers, barriers and value creation in the blue economy. Sustainable Technology and Entrepreneurship, 100018.
Giaretta, E., & Chesini, G. (2021). The determinants of debt financing: The case of fintech start-ups. Journal of Innovation & Knowledge. 6(4), 268-279. DOI: 10.1016/j.jik.2021.10.001
Metallo, C., Agrifoglio, R., Briganti, P., Mercurio, L., & Ferrara, M. (2021). Entrepreneurial Behaviour and New Venture Creation: the Psychoanalytic Perspective. Journal of Innovation & Knowledge, 6(1), 35-42. https://10.1016/j.jik.2020.02.001
Some of the statements you make are entirely obvious and should be supported in the text by these specific references.
3) The question could be asked of whether this study is representative of other sectors in your country or in the world. Please explain this potential applicability to a general context.
4) The statistical treatment is acceptable.
5) At the end of the ´Conclusion´ section, the author should include clear statements as to where research should now go –.
6) Finally, when you submit the corrected version, please do check thoroughly, in order to avoid grammar, syntax or structure/presentation flaws. Make sure you retain a formal/academic-specific style of presenting your work throughout the text - (if necessary) please seek for professional English proofreading services or ask a native English-speaking colleague of yours in order to refine and improve the English in your paper.
Thank you for the opportunity to read the paper.
Author Response
Reviewer 2
|
1 |
Abstract: Authors should state their contribution in terms of issue problems solved or ameliorated, theory or policy dilemmas resolved, or the like. Abstract should offer at least one example of a theoretical or managerial implication that authors concluded after their work. Response: We thank the reviewer for this observation. We have modified the abstract accordingly. |
|
2 |
The introductory/opening section should communicate a little clearer the literature gaps, as well as the study's aims & objectives in order to facilitate the flow of the study. Response: We have updated and edited the abstract and introduction sections to clarify our aims for the paper. |
|
3 |
Additional references to recent & relevant empirical studies could increase the quality of the research paper and provide a much clearer message to the reader - these may help you building your discussion which needs to be extended. Add the following to your reference list: Andati, P., Majiwa, E., Ngigi, M., Mbeche, R., & Ateka, J. (2022). Determinants of Adoption of Climate Smart Agricultural Technologies among Potato Farmers in Kenya: Does entrepreneurial orientation play a role? Sustainable Technology and Entrepreneurship, 100017. Dijkstra, H., van Beukering, P., & Brouwer, R. (2022). Marine plastic entrepreneurship; Exploring drivers, barriers and value creation in the blue economy. Sustainable Technology and Entrepreneurship, 100018. Giaretta, E., & Chesini, G. (2021). The determinants of debt financing: The case of fintech start-ups. Journal of Innovation & Knowledge. 6(4), 268-279. DOI: 10.1016/j.jik.2021.10.001 Metallo, C., Agrifoglio, R., Briganti, P., Mercurio, L., & Ferrara, M. (2021). Entrepreneurial Behaviour and New Venture Creation: the Psychoanalytic Perspective. Journal of Innovation & Knowledge, 6(1), 35-42. https://10.1016/j.jik.2020.02.001 Some of the statements you make are entirely obvious and should be supported in the text by these specific references. Response: We thank the reviewer for these suggestions. However, we are not quite sure how to respond to the last criticism “you make are entirely obvious and should be supported in the text by these specific references”. It would be necessary to receive some concrete examples so that we understand the problematic. We have had a look at every single citation mentioned above, but we find that their integration would appear very opportunistic as they really do not relate to the topic at hand except that they are concerned with entrepreneurship or sustainability in the broadest sense. They are not even concerned with gender/ women and entrepreneurship (e.g. topics cover entrepreneurial orientation and adoption of innovative technologies - business models of entrepreneurs concerned with marine plastic – entrepreneurial finance (fintech) – new venture creation through psychanalytic analysis). To satisfy the reviewer we integrate Andati et al. 2022 into the section concerned with drivers of sustainable entrepreneurial action. |
|
4 |
The question could be asked of whether this study is representative of other sectors in your country or in the world. Please explain this potential applicability to a general context. Response: Thank you very much for this suggestion. Although we don’t refer to a specific sector, we were considering the sustainability orientations/awareness of female entrepreneurs to integrate green/sustainable practices into their business. We have not analyzed any specific sector, but we focus on entrepreneurial women (we define our sample in the Methodology section). As for the general context, sustainability is an intrinsic part of Saudi Arabia’s economic development plan, which foresees sustainability being integrated into any sector. However, we find that sustainability is ‘prescribed’ in a top-down approach by the government. For it to be properly adopted by entrepreneurs, people need to be aware of the notion of sustainability and the relevance for their own lives. We think it requires more of bottom-up processes/consideration where people are aware of sustainability and not just through a state economic vision. We refer to this in conclusion. |
|
5 |
At the end of the ´Conclusion´ section, the author should include clear statements as to where research should now go Response: We have clarified our statement as to future directions of research. |
|
6 |
Finally, when you submit the corrected version, please do check thoroughly, in order to avoid grammar, syntax or structure/presentation flaws. Make sure you retain a formal/academic-specific style of presenting your work throughout the text - (if necessary) please seek for professional English proofreading services or ask a native English-speaking colleague of yours in order to refine and improve the English in your paper. Response: The paper has again been checked by a professional copy editor and we hope it fulfills your expectations. |
Round 2
Reviewer 1 Report
I want to thank the authors for considering the comments provided. I believe that the present version is far more consistent than the former.
The present version needs some final changes and improvements in form and writing style. Please consider a deep reading to build shorter sentences with more direct argumentation to raise the scientific soundness of your work.
Also, consider professional proof of this version and a consistent effort to clarify the purpose of each paragraph.
Finally, revise the Conclusions section to highlight the contributions and the value added by the paper as well as the exact policy recommendations brought to light by the reasearch to promote this entrepreneurial niche.
Best of luck with your research!
Author Response
Thank you very much for your further comments, we have added a section to the Conclusion to try to further highlight our contributions and also proof-read the paper and adjusted it again.
We hope you are satisfied with the paper.
Reviewer 2 Report
Authors should include the references that I suggested previously:
Andati, P., Majiwa, E., Ngigi, M., Mbeche, R., & Ateka, J. (2022). Determinants of Adoption of Climate Smart Agricultural Technologies among Potato Farmers in Kenya: Does entrepreneurial orientation play a role? Sustainable Technology and Entrepreneurship, 100017.
Dijkstra, H., van Beukering, P., & Brouwer, R. (2022). Marine plastic entrepreneurship; Exploring drivers, barriers and value creation in the blue economy. Sustainable Technology and Entrepreneurship, 100018.
Giaretta, E., & Chesini, G. (2021). The determinants of debt financing: The case of fintech start-ups. Journal of Innovation & Knowledge. 6(4), 268-279. DOI: 10.1016/j.jik.2021.10.001
Metallo, C., Agrifoglio, R., Briganti, P., Mercurio, L., & Ferrara, M. (2021). Entrepreneurial Behaviour and New Venture Creation: the Psychoanalytic Perspective. Journal of Innovation & Knowledge, 6(1), 35-42. https://10.1016/j.jik.2020.02.001
Author Response
We refer to the MDPI email.
Round 3
Reviewer 2 Report
Authors should improve the manuscript with the three references that I suggested:
Dijkstra, H., van Beukering, P., & Brouwer, R. (2022). Marine plastic entrepreneurship; Exploring drivers, barriers and value creation in the blue economy. Sustainable Technology and Entrepreneurship, 100018.
Giaretta, E., & Chesini, G. (2021). The determinants of debt financing: The case of fintech start-ups. Journal of Innovation & Knowledge. 6(4), 268-279. DOI: 10.1016/j.jik.2021.10.001
Metallo, C., Agrifoglio, R., Briganti, P., Mercurio, L., & Ferrara, M. (2021). Entrepreneurial Behaviour and New Venture Creation: the Psychoanalytic Perspective. Journal of Innovation & Knowledge, 6(1), 35-42. https://10.1016/j.jik.2020.02.001
Author Response
This issue has now been resolved.